# Measuring and valuing spillover effects in caregivers and families: A scoping review

**Tho T. H. Dang** [1]*, **Angeli Tabinga**[1], **Hannah Beilby**[2], **Natalie Barker** [3],
**Luke R. Johnson**[4], **Haitham Tuffaha**[2], **Luke B. Connelly** [2,5], **Angela M. Maguire**[6]

**1** Gallipoli Medical Research, Greenslopes Private Hospital, Greenslopes, Queensland, Australia,
**2** Centre for the Business and Economics of Health, The University of Queensland, St Lucia, Queensland, Australia, **3** Herston Health Sciences Library, The University of Queensland, Herston, Queensland, Australia, **4** Faculty of Medicine, The University of Queensland, Herston, Queensland, Australia, **5** The University of Bologna, Departimento di Sociologia e Diritto dell'Economia, Bologna, Italy, **6** Department of Veterans' Affairs, Australian Government, Brisbane, Queensland, Australia

* dangt@gallipoliresearch.org.au

## Abstract

### Objectives

As healthcare increasingly relies on informal care for chronic and complex conditions, economic evaluations have expanded beyond patient outcomes to consider spillover effects on caregivers and families. This scoping review aimed to map existing measures and methods for assessing these effects and to identify potential mechanisms, mediators, and moderators to inform future survey design.

### Methods

We conducted a comprehensive search of four databases (PubMed, APA PsycInfo, CINAHL Complete, and EconLit) for English-language studies published from 2017 to 2025, including quantitative, qualitative, and mixed-methods research reporting monetary or non-monetary spillovers. Screening and study selection followed the Participants, Concept, Context framework and were reported according to PRISMA-ScR guidelines.

### Results

A total of 141 studies met the inclusion criteria. Incorporating caregiver and family spillovers could meaningfully alter cost-effectiveness estimates, but this practice remained inconsistent due to limited data and methodological variability. Comprehensive assessment of spillover effects benefited from combining generic, caregiver-specific, and disease-specific tools to capture both perceived and measurable impacts. Valuation of societal and economic spillovers, including informal care time, costs, productivity loss, and wellbeing impact, was influenced by methodological choices, caregiver and patient characteristics, and caregiving context, highlighting the need for flexible,

**Data availability statement:** All relevant data are provided in the manuscript and Supporting information files.

**Funding:** This work was supported by the Returned and Services League of Australia, Queensland Branch, through a research funding agreement with Gallipoli Medical Research. The funding organisation did not play any role in the study design, data collection and analysis, decision to publish, or preparation of the manuscript. At the time the research was conducted, authors TD, AT, LJ, and AM were employed by Gallipoli Medical Research, a registered not-for-profit charity. Funding provided under the agreement supported the conduct of the research, including personnel time and research-related resources. The scientific and operational contributions of these authors—such as study design, data analysis, and manuscript preparation—are explicitly detailed in the 'Author Contributions' section.

**Competing interests:** I have read the journal's policy and the authors of this manuscript have the following competing interests: The senior author (Maguire) led the research whilst employed as a Clinical Psychologist and Principal Research Fellow at Gallipoli Medical Research. She is currently employed as a Specialist Mental Health Advisor, Department of Veterans' Affairs (DVA), Australian Government. The DVA had no role in the design and conduct of the study; collection, management, analysis, and interpretation of the data; preparation, review, or approval of the manuscript; or the decision to submit the manuscript for publication.

context-sensitive approaches. Caregiver outcomes reflected the interplay of mediating factors (psychological, social, relational) and moderating influences (coping, spirituality, support systems, caregiving intensity). Subjective caregiver burden was pivotal, shaping and conditioning the effects of caregiving stressors on wellbeing.

## Conclusion

Establishing consensus on best-practice approaches for incorporating spillover effects in economic evaluations is needed to accurately quantify their impact on caregivers and families and to inform interventions that reduce caregiver burden.

---

## Introduction

The health economics literature has traditionally focused on methods for measuring and valuing the effects of health conditions and interventions on patient outcomes. As populations age and healthcare resources are increasingly directed towards managing chronic illness, more prominence has been given to the role and costs of informal care. Accordingly, the literature has expanded from a patient-centric view to consideration of how informal care affects caregivers, families, healthcare systems, and society as a whole [1–6].

There are two overarching ways in which a family member's health and wellbeing may be affected by a care recipient's condition [7,8]. First, family members may be meaningfully involved in the provision of informal care (i.e., caregiving effects or 'caring for') [7,8]. This includes assisting with (instrumental) activities of daily living (I-ADLs), and providing emotional support and/or supervision (i.e., surveillance). Second, family members may be affected through their social and emotional ties with the care recipient (i.e., family effects or 'caring about') [7,8]. Here, physical or emotional proximity to the care recipient (i.e., co-residence or the closeness of the dyadic relationship) is an important determinant of caregiver or family member outcomes [9,10]. Within a family system, primary caregivers are likely to experience both caregiving effects and family effects. Bobinac et al. (2011) demonstrated that caregiving and family effects are separable and independently associated with caregiver health and wellbeing [7]. Failing to disentangle family effects from caregiving effects in economic evaluations can lead to overestimation of the impact of caregiving, and may bias results in favour of certain care recipient subgroups (e.g., younger patients with severe illnesses) [8].

There has been a significant increase in attention to the measurement of these spillover effects, defined by Basu and Meltzer (2005) as the "different direct and indirect welfare effects to all family members including the patient" and their incorporation into cost-effectiveness analyses [5]. The substantial burden of illness on caregivers and families is now widely recognised, and efforts to quantify the magnitude of these spillover have emerged [11]. Methods and tools for valuing spillover effects have evolved beyond conventional health-related quality of life (HRQOL) measures to include a focus on valuing informal care independently of health effects to avoid

double counting [12], a comprehensive catalogue of caregiver utility values for use in quality-adjusted life year (QALY) calculations – the standard metric enabling comparability across health interventions in the contemporary literature [13], and an algorithm to estimate caregiver time using patient-level EQ-5D data [14].

Understanding the mechanisms through which spillover effects arise, and the mediating and moderating factors that shape them, is critical for capturing the impact of illness beyond the patient. Mechanisms explain how a patient's health affects caregivers and family members, for example, through disrupted employment, emotional stress, or reduced social participation [15–17]. Mediators, such as caregiver burden or time spent on caregiving tasks, help identify the pathways through which these effects occur which can be targeted in interventions [18,19]. Moderators, including gender, socioeconomic status, and social support, influence the strength or direction of spillover effects, highlighting which subgroups are most vulnerable [20,21]. Examining these elements not only improves the design of caregiver and dyadic surveys and measurement tools but also supports the development of tailored interventions and more accurate economic evaluations that capture the full societal impact of health conditions [22].

There is an ongoing need for clearer guidance on best practices for valuing informal care time and measuring spillover burden through health utility metrics [11,23], as well as for additional research to develop guidelines for incorporating spillover effects in economic evaluations [4]. This need is intensified by the limited understanding of the factors that shape these spillover effects [24,25]. In response, the current scoping review sought to map existing measures and methods for assessing spillover effects on caregivers and family members. A secondary objective was to identify potential mechanisms, mediators, and moderators to inform the design of future surveys in this emerging area.

## Methods

We followed the scoping review framework outlined in the JBI Manual for Evidence Synthesis [26] and reported the review using the Preferred Reporting Items for Systematic Reviews and Meta-Analyses extension for Scoping Reviews (PRISMA-ScR) [27] (refer to the Scoping Review Protocol in S1 File and the PRISMA-ScR checklist in S2 File).

### Search strategy

A comprehensive search strategy was developed and peer-reviewed in line with the extension to the PRISMA statement for reporting literature searches [27,28]. Studies indexed in four databases (PubMed, APA PsycInfo, CINAHL Complete, and EconLit) were searched. The search strategy (see S3 File) was initially developed for PubMed and subsequently adapted for the other databases in consultation with experienced health and medical research librarians.

The search combined terms covering three key concepts: (spillover effects in caregivers and families) AND (health economics studies, methods, analyses, direct and indirect elicitation techniques) AND (values, costs, measures, instruments, outcomes). The search strategy was developed to retrieve results relevant to the primary objective of this scoping review (i.e., to provide an overview of the methodologies that have been used to measure and value spillover effects in caregivers and families), with the secondary objective met through screening the search results for studies that focused on mechanisms, mediators or moderators of spillover effects in caregivers and families.

The database searches were last executed on 30 April 2025. The screening process was developed and iteratively refined by five authors (HB, AT, HT, LC, AM) following an extensive review of the literature. Two authors (AT, AM) independently screened all titles and abstracts, and applied a single, mutually exclusive tag that represented the primary focus of the study with respect to the selection criteria (described below). Study selection was then verified by a third author (TD). Citation searches of the included studies were conducted to identify additional eligible records.

### Selection criteria

The Participants, Concept, Context (PCC) framework [29] was used to specify the inclusion and exclusion criteria for the review. Studies that (1) focused on measures, methods, mechanisms, mediators, or moderators of spillover effects in

caregivers and family members; and (2) reported at least one type of spillover, either monetary (e.g., the financial costs of informal care) or non-monetary (e.g., HRQOL, wellbeing, productivity, educational or occupational outcomes, and labour or social participation) were eligible. Quantitative, qualitative, and mixed-methods studies published in English from 2017 to 2025 were considered to ensure a focus on current literature. Studies were excluded if they (i) focused solely on patients' outcomes, (ii) involved only non-familial caregivers, (iii) examined intervention efficacy, cost of illness, or burden of disease, (iv) focused exclusively on instrument development or tool validation, (v) were non–peer-reviewed articles, comments, editorials, letters, errata/corrigenda, or protocols, or (vi) were not available in full-text.

## Data extraction

The data charting process followed the best practice approach recommended by Lockwood et al. (2019) [30], as outlined in the JBI Manual for Evidence Synthesis [26]. First, standardised forms for title/abstract screening and full-text data extraction were developed by five authors (AT, HB, HT, LC, AM). Second, two authors (AT, HB) independently piloted the data extraction forms on approximately 5% of the records at each stage to calibrate the forms. Third, data extraction was performed by two authors (TD, AT), with conflicts resolved by a third author (AM). Finally, the extracted data was synthesised by one author (TD), and the accuracy and completeness of the synthesised data was checked by a second author (AT).

Extracted information included author names, publication year, objectives, country, study design, participants, care recipients' conditions, sample size, measures of spillover, valuation techniques of spillover, analytical frameworks and methods, and key findings. Data were then synthesised into five categories: (i) study characteristics; (ii) issues in evaluating and incorporating spillover effects (in economic evaluations); (iii) comparison of instruments for measuring spillover effects; (iv) methodological approaches for valuing spillover effects; and (v) mechanisms, mediators, or moderators of spillover effects. The results are presented according to these categories in the following section.

## Results

### Search results

Fig 1 illustrates the identification of the 141 studies included for full-text data extraction and synthesis. The database searches last executed on 30 April 2025 retrieved 5039 records. After removing duplicates (n = 1433) and studies published prior to 2017 (n = 1802), 1804 records remained and were uploaded to Covidence for screening against the selection criteria, resulting in 133 studies meeting the inclusion criteria. An additional eight studies were identified through citation searches of the included studies.

### Study characteristics

Of 141 studies, most examined issues in evaluating and incorporating spillover effects (44%). Smaller proportions examined instrument comparisons (18%) and methodological approaches to valuation (18%), while one-fifth explored mechanisms, mediators, or moderators of spillover effects. Most studies were conducted in Europe (37%) or in international/multi-country contexts (28%). The main populations studied were unpaid/informal caregivers (60%), followed by caregiver–care recipient dyads (16%). Methodologically, most studies were quantitative (57%), particularly cross-sectional-based (70%) and longitudinal-based (25%) analyses. Reviews were common (33%), while mixed-methods (5%) and qualitative (4%) studies were relatively rare (Table 1).

### Issues in evaluating and incorporating spillover effects in economic evaluations

**Analytical approaches to evaluating spillover effects.** Studies used a range of approaches that revealed health and wellbeing spillover effects on caregivers and family members. Panel and survey-based econometric models showed that

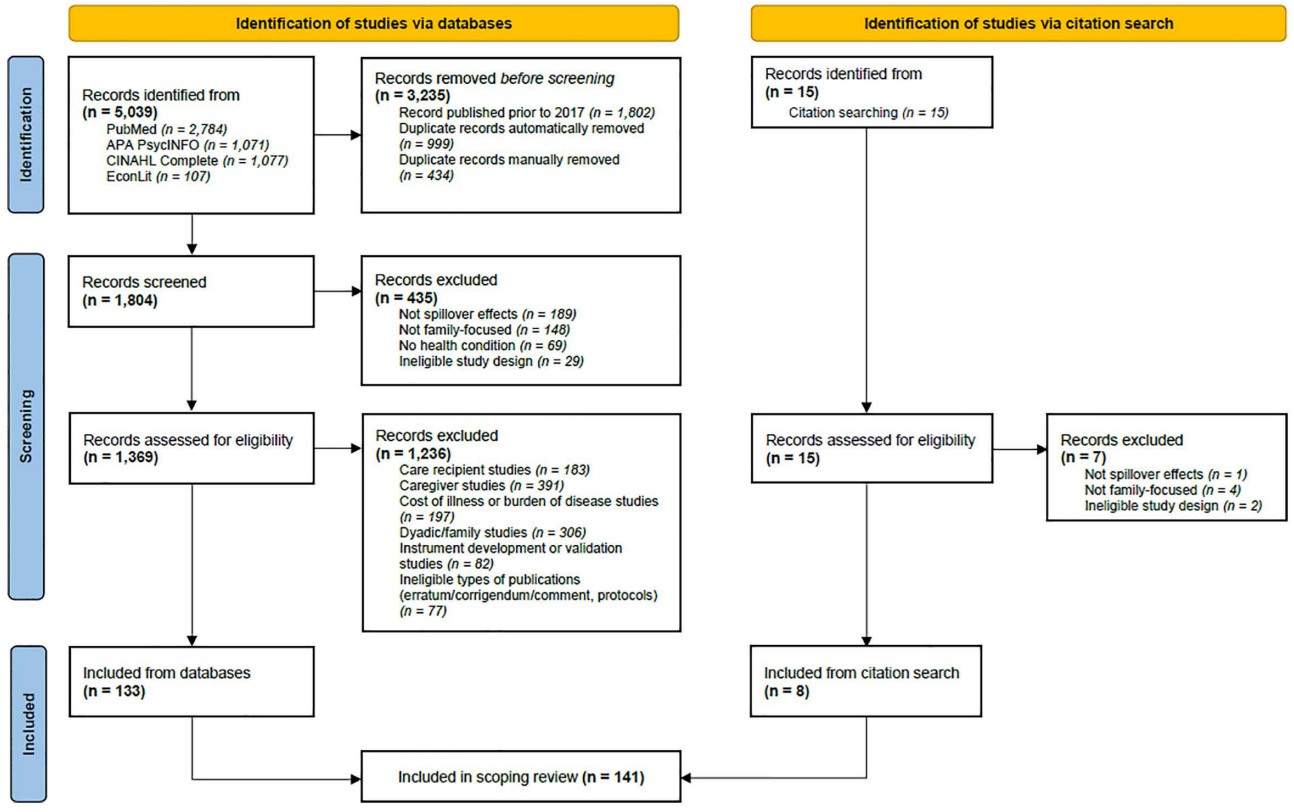

**Fig 1. PRISMA flow diagram of study identification.**

caregiving reduced mental health, especially for female caregivers and spouses [36]. Serious illness in a family member raised anxiety and depression among non-caregivers [51] and clinically meaningful QALY losses among relatives [59]. Quasi-experimental designs isolated causal effects, showing benefits of neonatal interventions on maternal mental health and siblings' education [41], and of older siblings' school-entry age on younger siblings' test scores [87]. Trial-based instrumental variable models revealed partner spillovers in smoking and alcohol treatments, which improved spousal outcomes and increased cost-effectiveness [48]. Clinical data confirmed strong parent–child HRQOL linkages, with parental wellbeing closely associated with child health status [85].

**Inclusion of spillover effects and broader societal value elements.** The studies highlighted the importance of considering family spillovers and broader value elements in health economic evaluations. When measured and included, caregiver HRQOL, impacts on familial psychological wellbeing, and societal costs (e.g., informal care, productivity losses, home modifications, and out-of-pocket caregiving expenses) could reduce incremental cost-effectiveness ratios (ICERs) and shift interventions across cost-effectiveness thresholds (Table S4.1 in S4 File). In most cases, the inclusion decreased ICERs, making interventions more cost-effective. For example, incorporating family health effects and informal care in analyses of Alzheimer's disease and paediatric interventions reduced ICERs by 31–42% and, in over one-third of cases, shifted values across commonly used cost-effectiveness thresholds [58,60,61,64,77,86]. In a smaller subset of studies, accounting for caregiver burden or household spillovers increased ICERs, reflecting situations where additional costs or reductions in caregiver quality of life (QOL) offset patient health gains [35,42,67]. Ignoring spillovers and focusing only on

**Table 1. Characteristics of studies on caregiver and family spillover effects (N = 141).**

| | Characteristics (n) | References |
|---|---|---|
| Primary focus | Issues in evaluating and incorporating spillover effects (n = 62) | [4,12,13,23,31–88] |
| | Comparison of instruments for measuring spillover effects (n = 25) | [89–113] |
| | Methodological approaches for valuing spillover effects (n = 26) | [114–139] |
| | Mechanisms, mediators, or moderators of spillover effects (n = 28) | [16,18,140–165] |
| Country | Europe (n = 52) | [16,31,32,35–37,39,41,42,50,51,53,55,72–74,76,82,83,89,97,98,103,104,107,112,114,116–120,122–125,127,129,131,133–136,138,139,144,146,149,152–154,160] |
| | Multiple countries/International (e.g., reviews) (n = 39) | [4,12,13,33,34,38,43–47,49,52,56–58,61,65–70,75,77,78,81,86,88,92,93,95,100,105,108,111,1-15,143,162] |
| | North America (USA and Canada) (n = 24) | [23,48,59,60,62,64,71,79,84,87,90,101,106,109,110,128,132,137,141,151,155,158,159,163] |
| | Other (Africa, Asia, Australia, and South America) (n = 25) | [18,40,54,63,80,85,91,94,96,99,102,113,121,126,130,142,145,147,148,150,156,157,161,164,165] |
| Studied population | Unpaid/informal carers (n = 84) | [4,13,16,18,32,33,36,38,40,42,43,45,49,50,53–55,58,60–65,67,68,71–74,80,81,83,84,86,88,-91–97,100,102–104,106–108,110–112,114–118,120,123,124,126,128–130,132–135,138,139,141,142,145,147,148,150–152,156,158,159,163,164] |
| | Care recipient – caregiver dyads (n = 22) | [31,34,41,56,76,82,85,90,98,101,113,119,121,122,127,136,140,155,160–162,165] |
| | Family members (in general or non-caregiving, including children) (n = 16) | [12,35,39,48,51,57,59,87,89,109,137,144,149,153,154,157] |
| | Other: general public, stakeholders, clinicians, researchers, not specified (e.g., reviews) (n = 19) | [23,37,44,46,47,52,66,69,70,75,77–79,99,105,125,131,143,146] |
| Care recipient's condition | Multiple diseases, i.e., a mix of different conditions and general care for individuals with acute or chronic illness, old age, or disability (n = 61) | [4,12,13,16,23,32,37,39,42–44,46,49,51,52,55,56,60,62,63,65–70,72–75,79,81,82,84,87,91-,92,94,99–102,106,108,114,118,125,128,129,131,132,138,143,144,146,152,154–156,158,160] |
| | Mental disorders, e.g., depression, substance misuse, and dementia (n = 23) | [36,45,48,50,54,59,61,76,97,103,104,107,111,115,130,136,147,149,157,161–164] |
| | Nervous system diseases, e.g., Alzheimer's disease, multiple sclerosis, and traumatic brain injury (n = 16) | [18,34,71,78,89,95,96,105,110,112,123,127,134,142,148,165] |
| | Paediatric conditions or childhood illness, e.g., low birth weight, autism spectrum disorder, and intellectual disability (n = 12) | [38,41,53,57,58,85,90,93,113,122,126,137] |
| | Cancer (n = 12) | [31,83,88,98,116,119,139,140,145,150,151,159] |
| | Other, e.g., COPD, COVID-19, diabetes, HIV, rare diseases, and skin conditions (n = 17) | [33,35,40,47,64,77,80,86,109,117,120,121,124,133,135,141,153] |

*(Continued)*

**Table 1.** (Continued)

| | Characteristics (n) | References |
|---|---|---|
| Study design | Quantitative (n = 81), including: | |
| | Cross-sectional-based analyses (n = 57) | [18,50,51,59,63,72,82–85,87,90,91,94,96–99,101,102,106,112,114,116–118,120,121,123,125–127,-130–133,135–138,140–142,144,145,147–150,156–158,160,161,163–165] |
| | Longitudinal-based analyses (n = 20) | [31,36,41,42,48,53,74,89,103,107,113,128,129,146,151–153,155,159] |
| | Economic evaluations (n = 4) | [35,64,76,122] |
| | Reviews, e.g., conceptual, literature, narrative, scoping, systematic reviews (n = 46) | [12,13,33,34,37,38,40,43–46,49,52,55–58,60,61,65–69,71,73,75,77–81,86,88,92,93,95,100,10-5,108,109,111,115,143,154,162] |
| | Mixed-methods (n = 7) | [4,54,62,70,124,134,139] |
| | Qualitative (n = 6) | [16,23,32,39,47,104] |

patient outcomes underestimated the societal burden of conditions such as cancer, dementia, and other chronic illnesses, as well as the full estimated value of interventions [31,34,49,56,57,60,62,71].

**Methodological challenges in incorporating spillover effects.** Despite recognition of these effects, inclusion of caregiver/family spillovers and broader value elements in economic evaluations remained inconsistent. Most studies acknowledged spillovers but only a few quantified or modelled them [13,56,57,60,61,66,73]. Health technology assessment guidelines rarely recommended their inclusion in the base case [37,38,56,57]. Some studies discussed concerns about definitional ambiguity and the scope of spillover effects, which limited comparability across evaluations [66,68]. In response, the Spillovers in Health Economic Evaluation and Research (SHEER) task force recently proposed working definitions of family, caregivers, and family and caregiver health spillovers for use in the context of cost-effectiveness analysis [4] and encouraged their adoption to support the development of good practice. Nevertheless, methodological barriers to capturing and incorporating spillover effects persist, which are highlighted as follows.

**Measurement of informal care time and costs remained inconsistent**. Most studies relied on recall-based methods such as interviews or questionnaires, with fewer using descriptive costing such as diaries or registry data [12,40], rather than rigorous incremental approaches, leading to wide variation in estimates, from as low as $30 to over $80,000 annually, even within the same disease area [65]. Time was reported as total hours or disaggregated by ADLs, IADLs, domestic tasks, community participation, leisure, or work [40,50]. Challenges included separating supervision from active care, reconciling caregiver versus recipient reports, and deciding which family members to include in analyses [39,54].

**The tools used to capture caregiver HRQOL and burden were not always well-suited**. While instruments such as EQ-5D and Zarit Burden Interview (ZBI) were commonly applied, they failed to capture key aspects such as stigma, memory loss, or emotional strain, particularly in conditions like dementia [54]. Many scales were not validated in certain populations, such as caregivers of cancer patients [54,88]. Mapping algorithms or 'crosswalks' that translate care- or disease-specific health outcomes into utility values [166] were developed to predict caregiver hours from care recipient preference-based HRQOL values [60]. However, crosswalks for estimating caregiver or family utilities from care- or disease-specific QOL instruments remain underdeveloped [55].

**Integration of family/caregiver spillover effects into QALY-based framework presented both technical and normative difficulties**. The framework typically overlooks non-health impacts such as empowerment, hope, and distributional effects [46,47,75] and carers' values [67]. Analysts faced choices about aggregating patient and caregiver utilities, applying multipliers, or presenting separate analyses, with risks of double counting or inequity [56,57,70]. Including spillovers could also bias resource allocation toward patients with larger family networks, raising equity concerns [52].

**Data limitations and reliance on precedent were repeatedly flagged as barriers**. Analysts often fell back on what had been done historically, or omitted spillovers due to a lack of high-quality data, despite growing recognition of their

importance [23,62]. Calls for methodological improvement included broader outcome [46,47], transparent reporting [23,37,52,69], use of modified impact inventory tables [23], equity-sensitive approaches [43,47,70], and longer time horizons to capture persistent effects [4,32,54,66,68,75,80].

In summary, caregiver and family spillovers could meaningfully shift cost-effectiveness estimates, yet their incorporation into economic evaluations remained inconsistent due to limited data and methodological variability, underscoring the need for standardisation and wider adoption in health technology assessments.

## Comparison of instruments for measuring spillover effects

**Carer-specific measures.** Carer-specific instruments, used to measure caregivers' HRQOL/wellbeing [91,94,98,99,102,108] and burden [93,97,100,101,111], were compared **psychometrically and clinimetrically**. These included generic care-related QOL tools (e.g., Adult Social Care Outcomes Toolkit for Carer [ASCOT-Carer], Care-related Quality of Life [CarerQOL]) [91,94,102], condition-specific care-related QOL measures (e.g., CareGiver Oncology Quality of Life [CarGOQOL], Schedule for the Evaluation of Individual Quality of Life – Direct Weighting [SEIQOL-DW]) [98,100,108], and carer burden scales (e.g., ZBI, Caregiver Reaction Assessment [CRA]) [97,100,101,111] (Table S4.2 in S4 File). Reliability was generally high (Cronbach's α > 0.80) with acceptable test–retest results [93,97,99,102,108,111]. Construct and content validity showed that ASCOT-Carer, CarerQOL, and Work–Family Enrichment Scales captured overlapping but distinct aspects, including HRQOL, burden, enrichment, and proxy versus self-reported impacts [91,94,98,100,101]. Convergent and discriminative validity were moderate to strong [97,102,108,111].

Advanced methods such as factor analysis [99,101] and item response theory [101] supported dimensionality and item performance. Responsiveness and sensitivity to change were reported for some burden and condition-specific care-related tools, including the ZBI, CRA, and CarGOQOL [93,97,108], though longitudinal validation was limited. Practical considerations, including respondent burden and ease of administration, were tested, with most instruments being feasible and quick to complete, although some newer or condition-specific tools required more time or guidance for respondents [102,108,111]. Reviews highlighted gaps in cultural adaptation, responsiveness, and coverage of different caregiving contexts, particularly beyond dementia, cancer, and dermatology [93,100,108,111].

**Utility measures.** The included studies used various instruments to elicit respondents' health-state preferences (hereinafter denoted as 'utility measures'). The outcomes from these techniques are referred to as 'utilities' when choices between health states involve uncertainty; and 'values' when they involve certainty [167]. These utilities and values can be used to generate the weights required to compute QALYs. Note that some instruments do not generate such weights, but generate rating-scale-type data (e.g., from a visual analogue scale [VAS]) that are neither utilities nor values of the type described above.

Utility measures were compared with other utility measures in **their ability to capture** caregiver HRQOL/wellbeing [89,90,106,113] and burden [89,113], while carer-specific instruments were compared in **their ability to assess** caregiver HRQOL/wellbeing [96,104,107], burden [96,107], and informal care time [107]. EQ-5D (3L or 5L), the most commonly used HRQOL tool, was less sensitive to social, emotional, and work-related caregiving impacts [89,90,113] and responded more to patient health changes [89], whereas SF-6D better captured social, behavioural [90], and caregiving effects [89]. The newer EQ Health and Wellbeing Short (EQ-HWB-S) instrument identified wellbeing and emotional spillover, distinguishing caregivers from non-caregivers and capturing differences by caregiver burden and care recipient condition [106,113].

While utility measures often overlooked caregiver burden and care time [107], carer-specific tools (e.g., CarerQOL-7D, ASCOT-Carer, Carer Experience Survey [CES]) were more sensitive to caregiving impacts, including patient health and hours of care and were generally preferred by participants [96,103,104,112]. Therefore, relying solely on utility measures may underestimate the full impact of caregiving [107] (Table S4.2 in S4 File).

**Disease-specific measures.** Studies comparing disease-specific instruments revealed **important insights into caregiver experiences and noted challenges in different contexts** [92,105,109,110]. The main issues identified

included: (1) the unclear concept of QOL, with variation in tools depending on who reported (self or proxy), what was measured, and where and for whom it was applied [92]; (2) few tools were specifically developed for a specific condition, with most adapted from broader populations having a spectrum of diseases, and only some underwent modern psychometric analyses (e.g., Rasch analysis) to assess suitability for different condition types or age groups [105], and (3) no disease-specific, accepted standard instrument existed for measuring caregiver QOL and burden [92,105,109] (Table S4.2 in S4 File). Gaps in standardisation and validation were emphasised. Authors noted the value of using multiple perspectives to reduce bias [92,110], and called for psychometric rigor and the use disease-specific tools [105,109].

Generic HRQOL tools (EQ-5D, SF-36/6D) complemented disease-specific measures across caregiver HRQOL/wellbeing [95,103,112] and burden [95,112], with disease-specific tools providing clinical detail and generic tools enabling comparisons across interventions [95]. Overall, a full assessment of caregiver spillovers benefited from combining generic, caregiver-specific, and disease-specific tools, and, when possible, adding novel technologies for objective monitoring [110] to capture both perceived and measurable spillover impacts on caregivers and family members.

## Methodological approaches for valuing spillover effects

Studies comparing valuation approaches highlighted systematic differences in methods used to value four domains of spillover effects of informal care: (1) informal care time, (2) informal care costs, (3) productivity losses, and (4) caregiver HRQOL/wellbeing.

**Informal care time.** Informal care time was most commonly valued using **replacement cost** (market wage of a professional caregiver or home help service) or **opportunity cost** (forgone earnings or leisure time) approaches [114–116,118–120,123], with a few studies combining both [40,138]. Other studies applied **contingent valuation** methods to elicit caregivers' willingness to pay (WTP) and/or willingness to accept (WTA) compensation for time spent or reduced caregiving [118,120,121,123].

Comparisons across methods indicated that opportunity cost often, but not always, produced higher estimates than the replacement cost, particularly for caregivers who reduced paid work or left the labour force [114,116,117,119,120,123], with variation arising from wage sources, task categories, leisure time valuation, and caps on caregiving hours [40,62,114,118]. The replacement cost provided a standardised and easily applied benchmark (i.e., wages of a paid professional), it however often underrepresented contributions from unpaid or non-working caregivers and the full time cost of care, with cross-country differences largely driven by the hourly value assigned by the local market rather than the intrinsic value of caregiving time itself [117,119]. In addition, it was limited by its unrealistic assumption of perfect substitutability with professional care [12,127] and less sensitive to variations in caregiver burden, patient age, and household context [127]. Contingent valuation approaches generally returned the lowest estimates. For example, in inflation-adjusted USD-2024, Engel et al. (2021) reported hourly caregiving costs of $22.9 (replacement cost), $18.7 (opportunity cost), and $17.4 (WTP via contingent valuation) [115]. Contingent valuation estimates could reflect caregivers' subjective perceptions but are sensitive to zero or protest responses and ethical considerations [120,121].

**Discrete choice experiments** (DCEs) were increasingly applied to quantify caregivers' WTA compensation for their time, capturing relative preferences across multiple dimensions of caregiving. DCE-based WTA estimates generally ranged from $9.2 per hour [131] to $17.4 per hour [130]. Studies applying the DCE method consistently found that valuations of informal care were highly dependent on type of caregiving task and caregiving intensity, and heterogeneity in caregiver preferences, including perceived impact of care [126,128,130,131]. For example, social or emotional support tasks tended to be valued higher, whereas routine household tasks were valued lower or even negatively [126,131]. Light caregiving (i.e., < 1,000 hours of care over a 2-year period) [128] sometimes enhanced caregiver wellbeing, whereas intensive caregiving (e.g., > 1,000 hours of care or caring for cognitive impairments) tended to reduce wellbeing significantly [128,130].

The DCE values thus reflected both the positive and negative mental, physical, relational, and financial effects associated with care [130,131]. These findings suggested that conventional wage-based methods (such as opportunity cost or replacement cost) likely underestimated the full societal and wellbeing costs of informal caregiving, sometimes by substantial margins, as they failed to capture those broader effects [126,128,130,131].

**Informal care costs.** Valuation of informal care costs extended beyond caregiving time to capture broader economic and societal contributions, and **the choice of valuation method substantially influenced estimated costs**. For example, annual informal care costs per caregiver of individuals with Alzheimer's disease ranged from $42,656–$49,971 (WTA via DCE) to $96,327–$113,482 (replacement cost) [123]. Even when the same valuation method was applied to caregivers of patients with similar conditions, estimates varied according to patient needs, caregiving intensity, caregiver employment status, and care setting (home, inpatient, post-acute) [84,117,121,122]. For instance, mean annual informal care costs for cancer caregiving estimated using the opportunity cost were $28,495 in Hanly et al. (2017) [116] and $26,312 in Oliva-Moreno et al. (2018) [168], whereas replacement cost-based estimates ranged from $18,244–$19,624 [116] to $22,515–$59,696 [168].

Translating care hours into economic outcomes was further complicated by **the risk of double counting** when valuation methods overlapped with QOL measures, particularly in studies using contingent valuation or conjoint analysis [12]. Estimates also varied over time, meaning total hours could obscure important variations in cost burden [84,119]. These discrepancies were especially pronounced when projecting population-level economic impacts of unpaid care [169].

**Productivity loss.** Productivity losses represented a substantial component of the societal burden of informal care [170]. Temporary cessation of work was consistently identified as a major driver of indirect costs, accounting for 12–17% of the total societal burden, while presenteeism added an additional 6–8% [114]. Productivity losses were treated as indirect costs and valued using either the **human capital approach** (estimating lost earnings up to retirement) or the **friction cost method** (limited to the period required to replace the absent worker) [33,69,77,86,114]. Applying friction cost instead of human capital reduced productivity estimates, particularly by excluding long-term absences [117]. Some studies embed productivity effects within opportunity cost measures, reflecting time diverted from paid work or education [132,169]. Methodological differences, including the choice of valuation approach and whether absenteeism or presenteeism was included, contributed to wide variability in estimates and underscored the importance of explicitly accounting for productivity effects in societal-perspective evaluations [45,69,70].

**HRQOL and broader wellbeing impacts.** To quantify preference-based estimations of HRQOL in caregivers, the **time trade-off** (TTO) method was commonly used to elicit utility weights for health state valuation [124,133–135,137,139]. TTO was frequently applied alongside VAS and structured health state vignettes, which were developed from literature reviews [124,134], clinical trial data [124,139], and qualitative interviews with patients, caregivers, and clinical experts [124,134,135,137,139]. Caregiver utilities varied with caregiving intensity and patient health, ranging from lower values during high-burden periods to higher values when patient health improved, as observed in seizure disorders and end-of-life scenarios (Table S4.3 in S4 File), reflecting the dynamic nature of caregiver spillover effects across the disease trajectory.

Al-Janabi et al. (2022) applied the **person trade-off** method to elicit public preferences for allocating health gains between patients and caregivers [125]. Most respondents (84%) traded between patient and caregiver HRQOL, with 42% prioritising patients, 19% prioritising caregivers, and 22% valuing both equally. Caregiver HRQOL was valued at between 0.69 and 0.74 relative to 1 for patient HRQOL, providing a basis for valuing different carer QOL outcomes in economic evaluations.

These findings highlighted that methodological choices substantially shaped estimates of the societal and economic value of informal care. Estimates were influenced by caregiver and patient characteristics, as well as the caregiving context, including both family and formal care settings, underscoring the need for context-sensitive and flexible approaches in economic evaluations [114,117,118].

## Mechanisms, mediators and moderators of spillover effects

**Mechanisms.** Spillover effects of caregiving and interventions operated through multiple mechanisms that influenced the health, wellbeing, and burden on caregivers, family members, and households (Table 2). These occurred through interconnected informational, behavioural, physiological, resource, and structural pathways across individual, dyadic, household, and societal levels.

**Informational** pathways involved knowledge transfer, such as how services informed and trained carers [16], as well as household-level information effects that shaped health behaviours and decision-making [143]. **Behavioural and**

**Table 2. Mechanisms underlying spillover effects.**

| Study | Care recipient condition | Spillover domain | Identified mechanisms |
|---|---|---|---|
| Al-Janabi-2019 [16] | Multiple diseases (dementia, stroke, and mental health) | HRQOL/Wellbeing impact | Six overarching mechanisms, through which health and social care services affect family carers' wellbeing, were identified. Each mechanism could have both positive and negative impacts on carers' wellbeing. They included: 1) Information: How service delivery informs and trains family carers. 2) Management of care: Shifts of responsibility for care between formal and family sectors. 3) Patient outcomes: Services changing patient outcomes. 4) Alienation: Feelings of alienation or inclusion created by service delivery. 5) Compliance: Barriers to patients complying and engaging with services. 6) Timing or location: Changes in the timing or location of services. |
| Benjamin-Chung-2017 [143] | Multiple diseases (infectious diseases and non-communicable diseases) | Family effect Various health outcomes (e.g., incidence of disease, nutritional status, mortality, health behaviours, unintended adverse consequences) | Six main mechanisms through which spillover effects occur identified: 1) Direct biological effects: e.g., reduced transmission of infectious diseases. 2) Direct resource effects: e.g., changes in the availability of health resources for others. 3) Behavioural effects: e.g., changes in health behaviours due to the intervention or perceived risk. 4) Market effects: e.g., changes in prices or availability of goods/services. 5) Information effects: e.g., increased knowledge or awareness. 6) Externalities related to social organization: e.g., changes in social networks or community norms. |
| Montoro-Gurich-2019 [154] | Multiple diseases (old age and chronically ill) | Caregiver burden HRQOL/Wellbeing impact | Economic crises can lead to a withdrawal of state responsibility for care, shifting the burden onto families, particularly women. This mechanism increases pressure on families to provide informal care, impacting their health and intergenerational solidarity. |
| Wuttke-Linnemann-2019 [162] | Dementia, Alzheimer's disease | Caregiver burden HRQOL/Wellbeing impact | Stress in dementia dyads is co-regulated. Mechanisms include: 1) Biopsychological mechanisms: Theories suggest that close relationships benefit health by regulating allostatic systems (cardiovascular, neuroendocrine, immune systems). 2) Stress-buffering model: Marriage reduces biopsychological stress, preventing brain changes that may lead to physical or mental health disorders. 3) Physiological co-regulation: Partners influence each other's physiological states, promoting emotional balance and health—especially in positive relationships. |

HRQOL, Health-related quality of life.

**care-responsibility** pathways captured how caregiving duties, patient adherence, and patient outcomes affected families, including broader shifts in care responsibilities between formal and informal sectors during austerity, which could exacerbate caregiver strain and reshape family dynamics [16,154]. **Physiological and dyadic** mechanisms highlighted stress co-regulation in dementia care, where biopsychological and stress-buffering processes influenced both patients and caregivers. Relationship quality was also critical, though most studies relied on caregiver reports rather than dyadic or biomarker-based measures [162]. **Resource and structural** mechanisms encompassed household resources and market effects [143].

**Mediating and moderating factors of spillover effects.** Caregiver outcomes, such as burden and HRQOL/wellbeing, were shaped by patient characteristics (e.g., frailty, comorbidities, depressive symptoms), caregiving context (e.g., demands, socioeconomic factors), and relational factors. These relationships were explained by a set of mediating variables, while also being contingent on moderating variables that buffered or exacerbated effects (Tables S4.4 and S4.5 in S4 File).

**Subjective caregiver burden** consistently emerged as a pivotal variable, functioning as both a mediator and a moderator. Burden mediated the links between patient symptoms (frailty, comorbidities, and depression), caregiving demands, and caregiver factors (income, education, and self-rated health) with psychological outcomes (e.g., anxiety, hopelessness, and HRQOL) [18,140,150,160,164]. Burden also moderated the impact of coping strategies on caregiver anxiety, amplifying the effects of maladaptive strategies such as denial, venting, and self-blame, while strengthening the protective role of adaptive strategies such as acceptance and positive reframing [152]. Its role varied across caregiver groups: for informal caregivers, burden was primarily an outcome linked to stress, physical strain, and care time, whereas for formal caregivers, burden acted as a moderator, with longer care time strengthening the association between care attitudes and experienced burden, suggesting that the effect of attitudes on burden was contingent on caregiving intensity [156].

**Social and relational factors,** including family functioning, spousal relationship quality, and social support, emerged as both mediators and moderators of caregiver outcomes. In adolescents, instrumental parentification mediated the relationship between caregiving roles and school achievement via general QOL [144]. Across adult caregivers, perceived social support and family functioning, mediated the impact of patient HRQOL and caregiving burden on family burden and depressive symptoms [145,147,161,164]. Relational supports, including strong family and spousal relationships, buffered against rising burden and mitigated the negative impact of caregiving on QOL [150,159]. Social support, encompassing network size, support receipt, satisfaction with support, and access to home- and community-based services, consistently enhanced resilience and reduced the effects of caregiving intensity on burden [142,163,165].

**Psychosocial and behavioural factors** operated as both moderators and mediators. As moderators, spirituality buffered the link between caregiver burden and depression, particularly under financial, scheduling, or family-support stressors, highlighting its role as a coping resource [151]. Coping strategies, both engagement- and tolerance-based, shaped stress–strain pathways and moderated the relationship between family stress and psychological symptoms [157]. As mediators, psychological constructs, including self-efficacy [141], hope [142,147], resilience [148], and mental health symptoms (anxiety and depression) [158], mediated the effects of stressors (including disease severity) and caregiving demands on HRQOL/wellbeing, and positive caregiving experiences.

Taken together, caregiver outcomes emerged from the interplay of mediating processes (psychological, social, relational factors) and moderating influences (coping, spirituality, support systems, caregiving intensity). Caregiver burden was especially pivotal, functioning at both levels to channel and condition the effects of caregiving stressors on wellbeing.

## Discussion

We reviewed 141 studies on evaluating and incorporating spillover effects, comparing measurement instruments, valuing spillovers, and understanding their mechanisms, mediators, and moderators. The included studies consistently indicated that ignoring spillovers substantially underestimated societal burden of diseases, particularly in chronic illnesses.

Incorporation of spillover effects into economic evaluations remained inconsistent due to methodological variability, limited data, and challenges in valuing informal care, productivity losses, and HRQOL impacts. Combining carer-specific, generic, and disease-specific instruments better captured perceived and measurable spillovers, while conventional wage-based methods often underestimated broader societal and wellbeing costs. Mechanistically, spillovers operated through inter-connected informational, behavioural, physiological, resource, and relational pathways, with caregiver burden as a key mediator and moderator across contexts.

The challenges in measuring and valuing spillover effects in caregivers and families reported in the literature may stem from various conceptual and practical problems. Persistent concerns around representativeness and generalisability were evident: many studies overrepresented spousal, female, middle-class, English-speaking, or hospital-recruited caregivers [124,128,130,151,164], while the inclusion of child carers, non-spousal caregivers, and complex caregiving networks remained minimal [161,163,164], restricting understanding of broader family spillovers. Dyadic and systemic perspectives were also underexplored, with few studies examining moderators, mediators, or bidirectional effects between caregivers and care recipients (e.g., Bannon et al. (2022) [34]; Sun et al. (2024) [159]; and Tsai et al. (2018) [18]), limiting insight into how caregiving impacts multiple family members over time.

Data quality and analytical approaches further compounded these challenges. Many studies relied on cross-sectional or short-term data [51,53,85,87,161,165], which limited causal inference and the ability to capture evolving spillover effects, including adaptation or coping mechanisms and delayed or cumulative impacts on physical and mental health. The reliance on self-report and proxy data introduced potential recall and endogeneity biases, particularly in retrospective reporting of caregiving time or HRQOL, and often failed to capture subjective or emotional nuances. Incomplete data on multi-recipient or non-household caregiving, care type, and post-death follow-up further reduced the precision of spillover estimates. Additionally, variability in care intensity, shared versus single caregiving arrangements, and cultural contexts complicated interpretation and limited generalisability. High heterogeneity in analytical aspects, including valuation methods, time horizons, and outcome instruments, reduced comparability across studies and limited opportunities for meta-analysis.

Recognition of broader value elements in economic evaluations has been increasing, but their incorporation into practice remained limited. Instruments often captured narrow caregiving dimensions (e.g., burden, HRQOL, ADLs, and financial constraints), lacked cultural adaptation, and omitted positive, relational, or coping outcomes, which might bias economic evaluations by overlooking benefits such as enhanced family cohesion, skill development, or emotional growth. Consideration of intergenerational, emotional, or reverse spillovers from caregiver to care recipient was also limited [36,51,53,66,85,120–122,124]. Other broader elements, such as cost savings outside the health system, reduction in uncertainty, value of hope, and health equity, were rarely incorporated despite their potential relevance [171].

Our scoping review makes several contributions to the extant literature. First, we update contemporary evidence, e.g., Grosse et al. (2019) [12] and Wittenberg et al. (2019) [13], on methods and measures for evaluating both the monetary and non-monetary costs of spillover effects in caregivers and families. Second, we integrate findings from a wide range of studies on spillover effects in caregivers and family members, irrespective of study design, analytical approach, care recipients' health conditions, or care settings, providing a broad understanding of how spillover effects have been measured and valued. Third, we detail key methodological and conceptual gaps and limitations, offering clear directions for future research. Fourth, we provide insights into the underlying mechanisms, as well as the mediating and moderating elements that connect independent factors to caregiver and family outcomes, representing a unique contribution of this review.

Nonetheless, there are limitations to this review. Our aim was not to appraise the quality of the included studies or assess their risk of bias, but rather to highlight key gaps and suggest directions for evaluating and incorporating spillover effects in familial surveys. Consequently, the included studies spanned a heterogeneous set of health conditions, precluding firm conclusions on the optimal approach for measuring and valuing spillover effects in any specific condition. Despite

this broad scope, it is notable that a substantial proportion (approximately 71%) of studies focused on caregivers and families of individuals with long-term health conditions associated with relatively high caregiver burden, such as chronic illnesses, aging-related care, and mental or neurodegenerative disorders. Finally, we excluded studies published in languages other than English, which may have introduced some bias into the review findings.

### Directions for future research

The paper highlights the need for outcome measures that combine generic, caregiver-specific, and disease-specific instruments to capture both perceived and measurable spillover impacts on caregivers and family members, adopting a societal perspective. Future research should include longitudinal studies to capture adaptation, coping mechanisms, and cumulative effects of caregiving over time. These studies should incorporate dyadic and network-level analyses to examine interactions between caregivers, care recipients, and other family members, including bidirectional spillovers and mediating/moderating factors. Triangulating multiple data sources, such as combining objective measures with self-report and proxy data and linking analyses to clinical trials, is recommended to reduce measurement biases, improve comparability, and enable meta-analyses. Developing unified frameworks for valuing and analysing spillover effects, with careful consideration of caregiving heterogeneity (e.g., ADLs vs. IADLs, shared vs. single caregivers, unpaid vs. leisure time), will enhance consistency across studies. Finally, research should address ethical, distributional, and equity considerations to ensure interventions reflect diverse caregiving experiences and broader societal priorities.

### Conclusion

Conceptual disagreements continue regarding the distributional consequences of including or excluding spillover effects in economic evaluations [4]. Methodological challenges remain in evaluating and explaining spillover effects in caregivers and family members. Given the factors driving the rising prevalence of spillover effects in families (e.g., increasing comorbidity, changes in family composition, withdrawal of state responsibility for care); routine measurement of spillover effects is required to improve the effectiveness and sustainability of health and social care systems, and to support access to institutional complements and substitutes for informal care.

### Supporting information

**S1 File. Scoping review protocol.**
(DOCX)

**S2 File. PRISMA-ScR checklist.**
(DOCX)

**S3 File. Search strategies.**
(DOCX)

**S4 File. Supplemental tables, including: Table S4.1. Impact on reported ICERs of including family/caregiver spillover effects and broader value elements.** Table S4.2. Psychometric and clinimetric comparison of instruments for measuring spillover effects. Table S4.3. Time Trade-Off studies for utility elicitation. Table S4.4. Mediators and moderators of spillover effects. Table S4.5. Factors acting as both mediators and moderators.
(DOCX)

**S1 Data. Minimal dataset.**
(DOCX)

## Acknowledgments

The authors would like to thank Lars Eriksson for peer reviewing the search strategy, and Greta Vos for updating the database searches in March 2023.

## Author contributions

**Conceptualization:** Angela M. Maguire.

**Data curation:** Tho T. H. Dang, Angeli Tabinga, Hannah Beilby, Natalie Barker, Angela M. Maguire.

**Formal analysis:** Tho T. H. Dang, Hannah Beilby, Haitham Tuffaha, Luke B. Connelly, Angela M. Maguire.

**Funding acquisition:** Haitham Tuffaha, Angela M. Maguire.

**Investigation:** Tho T. H. Dang, Angeli Tabinga, Angela M. Maguire.

**Methodology:** Tho T. H. Dang, Angeli Tabinga, Angela M. Maguire.

**Project administration:** Angeli Tabinga, Hannah Beilby, Natalie Barker, Angela M. Maguire.

**Supervision:** Luke R. Johnson, Haitham Tuffaha, Luke B. Connelly, Angela M. Maguire.

**Validation:** Tho T. H. Dang, Angeli Tabinga.

**Writing – original draft:** Tho T. H. Dang, Angeli Tabinga, Angela M. Maguire.

**Writing – review & editing:** Tho T. H. Dang, Angeli Tabinga, Hannah Beilby, Natalie Barker, Luke R. Johnson, Haitham Tuffaha, Luke B. Connelly, Angela M. Maguire.

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
