## [Decision Letter · Decision Letter 0]

5 Jan 2026

Measuring and valuing spillover effects in caregivers and families: A scoping review.

PONE-D-25-58821

Dear Dr. Dang,

We’re pleased to inform you that your manuscript has been judged scientifically suitable for publication and will be formally accepted for publication once it meets all outstanding technical requirements.

Kind regards,

Vlad Radoias

Academic Editor

PLOS One

“I have read the journal's policy and the authors of this manuscript have the following competing interests: The senior author (Maguire) led the research whilst employed as a Clinical Psychologist and Principal Research Fellow at Gallipoli Medical Research. She is currently employed as a Specialist Mental Health Advisor, Department of Veterans’ Affairs (DVA), Australian Government. The DVA had no role in the design and conduct of the study; collection, management, analysis, and interpretation of the data; preparation, review, or approval of the manuscript; or the decision to submit the manuscript for publication.”

We note that one or more of the authors are employed by a commercial company: Gallipoli Medical Research

Please respond by return email with an updated Funding Statement and Competing Interests Statement and we will change the online submission form on your behalf.

Additional Editor Comments (optional):

Dear authors,

based on the advice of one reviewer and on my own reading of the manuscript, I recommend your manuscript to be accepted for publication in PLoS ONE. Thank you for contributing to the journal.

Reviewers' comments:

Reviewer's Responses to Questions

**Comments to the Author**

1. Is the manuscript technically sound, and do the data support the conclusions?

Reviewer #1: Yes

2. Has the statistical analysis been performed appropriately and rigorously?

Reviewer #1: Yes

3. Have the authors made all data underlying the findings in their manuscript fully available?

Reviewer #1: Yes

4. Is the manuscript presented in an intelligible fashion and written in standard English?

Reviewer #1: Yes

Reviewer #1: This manuscript is a methodologically robust scoping review that makes a clear and substantive academic contribution to the fields of health economics and caregiving research. The focus on measuring spillover effects among caregivers and families is both critical and timely, particularly in the context of population aging and the growing burden of chronic disease. The review systematically synthesizes existing conceptual frameworks, measurement instruments, and methodological gaps in the current literature, presenting them clearly and coherently.

No concerns were identified regarding research ethics or publication ethics. As this study is based on a review of previously published literature, ethical approval is not applicable. There is no indication of dual publication or redundant reporting, and the sources are appropriately cited. Overall, the manuscript is of high quality, aligns well with the journal's scope, and is suitable for publication, with only minor refinement to the discussion of implications for future research and policy.

**Do you want your identity to be public for this peer review?** For information about this choice, including consent withdrawal, please see our Privacy Policy

Reviewer #1: No

---

## [Editor Report · Acceptance letter]

PONE-D-25-58821

PLOS One

Dear Dr. Dang,

I'm pleased to inform you that your manuscript has been deemed suitable for publication in PLOS One. Congratulations! Your manuscript is now being handed over to our production team.

Kind regards,

on behalf of

Dr. Vlad Radoias

Academic Editor

PLOS One